# Provision of Safe Anesthesia in Magnetic Resonance Environments: Degree of Compliance with International Guidelines in Saudi Arabia

**DOI:** 10.3390/healthcare11182508

**Published:** 2023-09-10

**Authors:** Mohammed S. Alshuhri, Bader A. Alkhateeb, Othman I. Alomair, Sami A. Alghamdi, Yahia A. Madkhali, Abdulrahman M. Altamimi, Yazeed I. Alashban, Meshal M. Alotaibi

**Affiliations:** 1Radiology and Medical Imaging Department, College of Applied Medical Sciences, Prince Sattam Bin Abdulaziz University, P.O. Box 422, Alkharj 11942, Saudi Arabia; m.alshuhri@psau.edu.sa; 2Radiology Department, King Salman Hospital, Cluster One Riyadh, Ministry of Health (MOH), Riyadh 12769, Saudi Arabia; baalkhateeb@moh.gov.sa; 3Radiological Sciences Department, College of Applied Medical Sciences, King Saud University, P.O. Box 145111, Riyadh 4545, Saudi Arabia; salghamdi1@ksu.edu.sa (S.A.A.); yalashban@ksu.edu.sa (Y.I.A.); mbakheet@ksu.edu.sa (M.M.A.); 4Department of Diagnostic Radiography Technology, College of Applied Medical Sciences, Jazan University, Jazan 45142, Saudi Arabia; ymedkhali@jazanu.edu.sa; 5King Faisal Specialist Hospital & Research Center, Riyadh 11564, Saudi Arabia; abdulraltamimi@kfshrc.edu.sa

**Keywords:** magnetic resonance imaging, safe anesthesia, Saudi Arabia

## Abstract

Background: The lack of local guidelines and regulations for the administration of anesthesia in magnetic resonance imaging (MRI) units presents a potential risk to patient safety in Saudi Arabia. Hence, this study aimed to evaluate the extent to which hospitals in Saudi Arabia follow international guidelines and recommendations for the safe and effective administration of anesthesia in an MRI environment. Methods: This study used a questionnaire that was distributed to 31 medical facilities in Saudi Arabia that provided anesthesia in MRI units. Results: The findings of the study revealed that the mean compliance with the 17 guidelines across the 31 sites was 77%; 5 of the 31 sites (16.1%) had a compliance rate of less than 50% with the recommended guidelines. Only 19.4% of the institutes provided general safety education. Communication breakdowns between anesthesia providers and MRI teams were reported. Conclusions: To conclude, this survey highlights the status of anesthesia standards in Saudi Arabian MRI units and emphasizes areas that require better adherence to international guidelines. The results call for targeted interventions, including the formulation of specific national anesthesia guidelines for MRI settings. Communication breakdowns between anesthesia providers and MRI teams were reported at a rate of 83.9% during the administration of a gadolinium contrast agent. There were additional breakdowns, particularly for high-risk patients with implants, such as impaired respirators (74.2%), thus requiring further investigation due to potential safety incidents during MRI procedures. While considering the limitations of this study, such as potential biases and the low response rate, it provides a valuable foundation for refining protocols and promoting standardized practices in Saudi Arabian healthcare.

## 1. Introduction

Magnetic resonance imaging (MRI) is a non-invasive diagnostic imaging modality that has revolutionized medical imaging because of its ability to provide high-quality images without the use of ionizing radiation. The increased availability and utilization of anesthesia services for MRI have played a crucial role in facilitating the imaging process, especially for pediatric patients and individuals who may struggle with remaining still during the procedure. By providing sedation or general anesthesia, healthcare professionals can ensure the comfort and safety of patients, thus enabling high-quality imaging results.

Emerging research highlights that a significant percentage of pediatric patients—ranging from 48% to 91%—require anesthesia during MRI procedures [1,2]. This escalating demand can be partly attributed to the expanding utilization of MRI surveillance for children with cancer and cancer predisposition syndromes. Furthermore, there has been a noticeable rise in the number of geriatric patients with dementia who require MRI examinations. These patients often require sedation or general anesthesia to ensure their cooperation and comfort during the procedure. The growing range of conditional monitoring devices that use MRI enhances access to MRI for individuals with pacemakers, cochlear implants, and other implantable devices, thus eliminating previous barriers and expanding the possibilities for accurate diagnostic imaging. This breakthrough enables previously ineligible patients who were unable to undergo MRI scans to now benefit from this diagnostic modality [1,2,3].

However, it is essential to recognize that administering anesthesia in an MRI environment presents unique challenges. MRI departments are characterized by the presence of strong magnetic fields, radiofrequency (RF) waves, excessive noise levels, and restricted access to patients during imaging studies. These factors create a distinct set of risks that need to be carefully managed to ensure patient safety and the smooth operation of the MRI procedure. Previous studies have shed light on the significance of MRI-related incident reports, which account for approximately 13% of all incident reports within radiology departments [3,4]. These incidents can include a range of issues, such as equipment malfunctions, patient misidentification, medication errors, and adverse events related to anesthesia administration.

In a study by Jaimes and colleagues [1], the utilization of sedation or general anesthesia during medical procedures was found to be associated with a significantly higher rate of safety reports. This association remained significant even after accounting for potential confounding factors, such as patient age and location. Specifically, the rate of safety reports in sedated patients was 0.8%, which was nearly twice as high as the rate of 0.45% observed in non-sedated patients. The specific reasons for this difference were not investigated in the study. However, the study suggested that factors such as the administration of anesthesia to inpatients and the involvement of multiple staff members who may be less familiar with MRI hazards and monitoring equipment could contribute to MRI safety-related hazards. The complex and dynamic nature of the MRI environment necessitates rigorous adherence to safety protocols, effective communication among healthcare providers, a requirement for experienced personnel, and ongoing monitoring. This comprehensive approach is essential for proactively preventing and rapidly addressing any potential risks or complications that might emerge [1,2,3,4,5,6].

To mitigate these risks, comprehensive guidelines and protocols for anesthesia administration in MRI settings have been developed to enhance patient safety and optimize the quality of imaging results. The Association of Anaesthetists in the UK published guidelines in 2002, with an update on the safety of MR units in 2019 [6,7,8,9]. Furthermore, the American Society of Anesthesiologists released a practice advisory providing specific recommendations for safe anesthesia administration in MRI environments [7,8,9,10]. These guidelines address factors such as patient selection, medication management, monitoring, emergency preparedness, and staff training and education. Adhering to these guidelines, along with implementing robust quality assurance and incident reporting systems, is crucial for promoting a culture of safety within MRI departments.

Despite the existence of national and international guidelines, the Ministry of Health in the Kingdom of Saudi Arabia currently lacks written guidelines or regulations specifically addressing the administration of anesthesia in MRI units [8]. As a consequence, local healthcare institutions often depend on policies and guidelines established by international organizations to guide their anesthesia practices in MRI settings. This lack of national guidelines and regulations presents potential risks to patient safety within Saudi Arabia. The absence of standardized protocols tailored to the specific needs and practices of the country may result in variations in anesthesia administration, inadequate safety measures, and suboptimal patient outcomes. Given the unique challenges associated with providing anesthesia in an MRI environment, it becomes crucial to establish comprehensive and locally adapted guidelines that ensure the safe and effective delivery of anesthesia services.

The primary objective of this study was to comprehensively assess the degree to which hospitals in Saudi Arabia align with international guidelines and recommendations for the safe and effective administration of anesthesia within MRI environments. Furthermore, the study examined the extent of collaboration between anesthesia departments and radiology departments. Through this multifaceted assessment, this study sought to provide a holistic understanding of the current state of anesthesia administration in MRI units in Saudi Arabia and identify areas for potential improvement.

## 2. Materials and Methods

The survey was cross-sectional by design, with an assessment period of five months (September 2022 to January 2023). A total of 58 medical facilities in Saudi Arabia that provide anesthesia services to MRI units were included in the survey. To collect the necessary data, a survey was designed and distributed to the participating institutes. The survey was shared through a link provided in an email, which was sent to the heads of the anesthesia department or the lead anesthetists responsible for administering anesthesia during MRI procedures. Confidentiality and anonymity options were provided to ensure the participants’ privacy. Although institutional review board approval was not sought for this voluntary survey, as it involved provider and hospital practices without any patient or protected health information, ethical considerations were upheld throughout the study. The initial invitation email was followed by a reminder email that was sent two weeks later to maximize the response rate. The survey questionnaire was divided into sections. The first part aimed to gather background and demographic information from the respondents, including details such as their role (e.g., consultant or non-consultant) and the type of institute in which they practiced (e.g., tertiary or non-tertiary hospital).

### 2.1. Guideline Compliance Assessment

The subsequent questions in the survey were designed to evaluate the degree of compliance with 15 guidelines, which were adopted from reputable sources, such as the Association of Anaesthetists (version 2019) and the American Society of Anesthesiologists (version 2015). In cases where the two documents conflicted, the more recent standard was adopted for consistency [6,7].

### 2.2. Risk Assessment

To measure the impact of non-compliance with the guidelines, a risk matrix based on Failure Mode and Effect Analysis (FMECA) was applied. For each considered element, a judgment on predefined criteria, built on a scale from 1 to 5, was associated with calculating a Priority Risk Number (PRN) [11]. A PRI is calculated as the product of three characteristics: the likelihood of the event occurring, the severity of the event’s impact, and the risk appetite [12].

### 2.3. Statistical Analysis

The data obtained from the survey responses were compiled into a spreadsheet to facilitate analysis. Numeric variables were quantified, and statistical analyses were conducted by using GraphPad Prism (version 7 for Windows, San Diego, CA, USA) as required to uncover patterns, trends, and any significant findings. To assess compliance levels across different hospital types, chi-square and Fisher exact tests were utilized. Additionally, independent-sample t-tests were executed to gauge the mean differences in compliance between tertiary and non-tertiary hospitals, as well as between university and non-university hospitals. All statistical tests were considered significant with *p*-values of less than 0.05.

## 3. Results

### 3.1. Compliance with International Guidelines

In this study, a total of 31 centers out of 58 responded to the survey, resulting in a response rate of 53%. Among these, twenty-four hospitals (77.4%) were located in Saudi Arabia’s major cities—Riyadh, Jeddah, and Makkah—while the remaining hospitals were spread across seven smaller cities with populations below one million based on the 2020 Saudi national statistics. Participant roles encompassed the following: 67.7% were consultants, 9.7% were registrars, 6.5% were senior registrars, 3.2% were residents, and 12.9% were sub-specialty consultants. The average professional experience of the respondents was 14.76 ± 2.43 years, with 74.2% having over a decade of experience. The participant demographics included 29% Saudis and 71% non-Saudis, with 93.5% male and 6.5% female participants.

The compliance rates of the hospitals were categorized into four groups based on their adherence level, which ranged from less than 50% to over 95%. Notably, three sites achieved a perfect compliance rate of 100%, while five sites (16.1%) fell below the 50% compliance threshold. Tertiary hospitals were compared with non-tertiary hospitals in terms of their degree of fulfillment of the guidelines. Both groups showed nearly identical levels of compliance (75.7% vs. 72.8%). Among the participating hospitals, university hospitals (consisting of nine sites) demonstrated a notably higher compliance rate of 94.12% than that of non-university hospitals, which had a compliance rate of 61.77%.

### 3.2. Present Anesthesia Practices and Safety Concerns in MRI Units

Table 1 displays the current anesthesia practices in MRI units. A total of 16 centers (52.4%) reported conducting more than 200 MRI examinations under anesthesia per year based on their data. Approximately 22.6% of the hospitals restricted the administration of sedation in remote locations exclusively to anesthesiologists, while 77.4% allowed radiologists to perform sedation. In terms of general anesthesia, around 54.8% of hospitals limited aesthesia administration in MRI units to consultants, while the remaining 45.2% permitted anesthesiologists of all levels to conduct general anesthesia. In the survey, 25.8% of the respondents expressed concerns regarding the safety procedures of MRI machines during anesthesia administration, while 74.2% had no issues. Of the participants, 16.1% reported accidents, such as hypoventilation, inadequate ventilation, lack of newborn monitoring, equipment failure, machine breakdown during the procedure, and sedation with inadequate monitoring and oximeter facilities, whereas 83.9% had no concerns. Regarding the labeling of anesthesia equipment, 71% of hospitals indicated that all anesthesia monitors in MRI units were correctly labeled according to the ACR standards (safe, conditional, and unsafe). Approximately 80% of the participants reported that patients with implanted devices underwent screening procedures before MR examinations, which encompassed MR screening forms, patient interviews, and measures for ensuring patient safety and precise device identification during imaging. Among the surveyed hospitals, 83.9% had established pre-procedural notification protocols. In this setup, the MRI department informed the designated anesthesiologist about specifics such as the MRI procedure, scan duration, use of contrast media, and type of MRI sequences. A total of 90.3% of the participants agreed that the MRI unit at their facility always had suction equipment available for quick access to the patient’s airway. The availability of alternative MRI-compatible airway devices for MRI suites in their facilities was agreed upon by 87.1%.

### 3.3. Training and Education

The results of the study on the conformity of the departments of anesthesiology to recommendations related to MRI safety training for anesthesiologists working in MRI units are presented in Table 2. Of the 31 institutions, 41.9% reported having a written policy or guideline for safety regulations to be followed when administering anesthesia in MRI units. However, the majority of the institutions (58.1%) reported that there were no such policies or guidelines for their facilities. Only 19.4% of the departments of anesthesiology reported having been provided with general safety education for their staff on the specific physical environment of the MRI scanner. In contrast, 80.6% of the departments reported that no general safety education was provided. In terms of emergency services, 45.2% of the departments of anesthesiology reported having trained their staff in handling cardiac arrests in MRI units, whereas 54.8% reported no such training.

### 3.4. Communication between the Anesthesia Provider and MRI Team

As presented in Table 3, the majority of the respondents reported a communication breakdown between the department of anesthesiology and the MRI team when treating patients with high-risk medical conditions and impaired respirators (74.2%). Similarly, 74.2% of the respondents reported a communication barrier with the MRI team in determining whether patients required physiological or invasive monitoring. Regarding screening for implanted devices, 77.4% of the respondents reported a communication barrier with the MRI team in determining whether the patients were screened for implanted devices, pacemakers, ferromagnetic items, or surgical clips. In addition, 83.9% of the respondents reported a communication barrier with the MRI team in determining whether patients would be administered a gadolinium contrast agent. Finally, with regard to communication between anesthetists and MRI units to determine whether patients had high-risk conditions and how they were managed, 77.4% of the respondents confirmed such communication.

### 3.5. University vs. Non-University Hospitals

The analysis involved a comparison between university and non-university hospitals in terms of their adherence to guidelines, as outlined in Table 4. The comparison indicated variations in the implementation of anesthesia-related guidelines between the two types of hospitals. While some guidelines showed similar proportions in both university and non-university hospitals, there were notable differences in safety education and specialized training for emergencies in MRI environments. Written policies/guidelines for safety regulations during anesthesia in MRI were found in 66.7% of the university hospitals but absent in the non-university hospitals (*p* = 0.001). University hospitals (88.9%) had a higher presence of pre-procedural notification protocols than non-university hospitals did (63.6%), but the difference was not statistically significant (*p* = 0.1). Similarly, university hospitals (88.9%) exhibited comprehensive knowledge about monitoring equipment, while this was observed in only 68.2% of non-university hospitals (*p* = 0.9). General safety education about MRI environments was provided for anesthesiologists in all university hospitals (100%) but only in 31.8% of the non-university hospitals (*p* = 0.001). Moreover, specialized training for MRI emergencies, including cardiac arrests, was significantly more prevalent in university hospitals (77.8%) than in in non-university hospitals, which demonstrated a prevalence of a mere 5% (*p* < 0.001). Communication practices encompassing aspects such as addressing high-risk patients, using monitors, managing implanted devices, using a gadolinium contrast agent, and handling high-risk conditions, were found to be similar between the two hospital types, with comparable proportions and non-significant *p*-values (all *p* > 0.9).

### 3.6. Tertiary vs. Non-Tertiary Hospitals

Comparing tertiary and non-tertiary hospitals revealed minor variations in the fulfillment of guidelines related to anesthesia practices (Table 5). While some guidelines exhibited similar proportions, no significant differences were observed in the safety education, specialized training, or communication practices between these two categories of hospitals. Tertiary hospitals had slightly higher sedation use (23.5% vs. 20%) and more general anesthesia (58.8% vs. 46.7%), but not significantly so (*p* > 0.9, *p* = 0.8). All tertiary hospitals had assistants in MRI units (100% vs. 92.9%), and MRI equipment labeling was in 82.3% of tertiary and 71.4% of non-tertiary hospitals (*p* = 0.9). Pre-procedural notification (88.2% vs. 64.2%) and monitoring knowledge (88.2% vs. 78.6%) were higher in tertiary hospitals, but not significantly so (*p* = 0.7, *p* > 0.9). Written safety guidelines for MRI anesthesia had a prevalence of 29.2% in tertiary hospitals and 20% in non-tertiary hospitals (*p* > 0.9). General safety education for MRI was in 53% of tertiary hospitals and 35.7% of non-tertiary hospitals, and specialized training for MRI emergencies was in 18.3% of tertiary hospitals and 20% of non-tertiary hospitals (*p* > 0.9). Communication was similar (all *p* > 0.7).

### 3.7. PRN Evaluations

The results of the risk prioritization assessment for various health hazards associated with anesthesia in MRI units are presented in Table 6. The risk prioritization indicates the level of risk associated with each health hazard, with higher PRN values signifying greater risk. The results highlight areas where attention and mitigation efforts should be focused to enhance patient safety in MRI environments. Notably, hazards such as “Absence of specialized training for MRI emergencies” and “General safety education about MRI environment” demonstrate the highest risk prioritization.

## 4. Discussion

The present statistical survey not only provides valuable insights into the anesthesia standards in Saudi Arabia, but also contributes to the overall evaluation and positioning of these standards, particularly in the context of administering anesthesia in MRI units. The findings emphasize the need for continuous efforts to improve compliance with recommendations to ensure optimal patient care and safety. Identifying areas of lower compliance can guide targeted interventions and quality improvement initiatives. The risk prioritization assessment underscores critical risks, including the absence of specialized MRI emergency training, communication breakdowns, and unclear anesthesia policies in MRI units. Consequently, targeted interventions like specialized training, improved communication protocols, and well-defined guidelines are imperative.

The considerable inter-site variation observed in the degree of compliance, which ranged from 41.8% to 100%, is a concerning finding. This variation suggests that there may be significant discrepancies in the quality and consistency of anesthesia care provided across different units and institutions. Addressing this variability is of paramount importance for promoting standardized and optimal anesthesia practices throughout Saudi Arabia [8,13]. The study’s findings also shed light on the compliance rates regarding guidelines for anesthesia administration in MRI environments, revealing interesting patterns among different types of hospitals. The comparison between university and non-university hospitals in terms of their adherence to anesthesia-related guidelines reveals distinct variations in the implementation of these guidelines. While some aspects showed similar proportions in both categories, noteworthy differences emerged in safety education and specialized training for emergencies within MRI environments. One significant finding was that of the presence of written policies for safety regulations during anesthesia in MRI, which was notably greater in university hospitals (66.7%) in comparison with their complete absence in non-university hospitals (*p* = 0.001). This underscores the commitment of university hospitals to well-defined safety protocols for anesthesia procedures in MRI settings. Additionally, the greater presence of pre-procedural notification protocols in university hospitals (88.9%) than in non-university hospitals (63.6%) suggests a proactive approach to ensuring adequate preparation before anesthesia procedures, although the difference was not statistically significant (*p* = 0.1). Interestingly, university hospitals demonstrated a significantly greater focus on safety education, including general education about MRI environments (100% vs. 31.8%, *p* = 0.001) and specialized training for MRI emergencies (77.8% vs. 5%, *p* < 0.001). The higher compliance rate observed in university hospitals highlights the potential influence of factors such as advanced educational programs, research-driven environments, and a multidisciplinary approach to healthcare [2,14].

Moving on to the comparison between tertiary and non-tertiary hospitals, minor differences were observed in the fulfillment of guidelines for anesthesia practice. Although some variations were noted, no statistically significant differences were found in safety education, specialized training, or communication practices between these two categories of hospitals. This suggests that the complexity of the hospital setting may not be the sole determinant of compliance with anesthesia guidelines in MRI environments [14].

Communication breakdowns between anesthesiologists and MRI teams can manifest in various aspects of patient care. In the context of this study, a noteworthy observation is that a majority of respondents, irrespective of their hospital types, reported encountering communication barriers with the MRI team. Previous studies have emphasized the importance of effective communication between anesthesia providers and MRI teams [11,13,15]. Mahmoud et al. (2013) found that communication breakdowns between anesthesia providers and MRI technologists can lead to errors in patient care, such as delays in treatment and adverse events [15]. Furthermore, the study found that a lack of communication and teamwork between the two groups could lead to poor patient outcomes. This could cause potential clinical risks for patients or health professionals [15]. They include but are not limited to minor to severe permanent injuries and, in extreme cases, the deaths of the patients. Effective communication between anesthesia providers and MRI teams is essential for ensuring patient safety and improving the quality of care. Strategies such as regular interdisciplinary meetings, checklists, and standardized protocols for patient screening and monitoring can improve communication and collaboration between these groups [15]. Therefore, it is important to establish training programs that emphasize the importance of effective communication and collaboration between these groups to ensure patient safety and improve the quality of care. Furthermore, ongoing quality improvement efforts should focus on identifying and addressing communication breakdowns in order to improve patient outcomes.

The findings of our study raise concerns about the low percentage of participants (19.4%) who received MRI safety training. This highlights the urgent need for medical facilities in Saudi Arabia to develop and implement specialized MRI safety training programs specifically designed for anesthetists. Building upon a previous study that revealed a 60% overall knowledge score among non-imaging healthcare workers in the country, it is crucial to address the identified knowledge gaps [16]. Although the study did not provide specific details about the participation of anesthetists, it is important to recognize their unique role and tailor educational initiatives to enhance their understanding and practice of MRI safety. Based on the risk prioritization assessment in this study, it is crucial to prioritize MRI safety training programs for anesthetists to ensure the highest safety standards during MRI-related anesthesia procedures. By providing targeted education and training, medical facilities can enhance patient care, mitigate potential risks, and uphold the highest standards of safety and quality in MRI examinations. These training programs should cover a range of topics, including MRI hazards, patient positioning, equipment safety, emergency protocols, and effective communication within the MRI team, as is recommended by the ACR guidelines [17]. Furthermore, ongoing assessment and continuous professional development initiatives should be implemented to ensure that anesthetists stay up to date with the latest advancements and best practices in MRI safety. Collaborative efforts among medical institutions, professional associations, and regulatory bodies are necessary to establish standardized guidelines and accreditation processes for MRI safety training programs tailored to anesthetists.

The international advisory suggests that it is crucial for anesthesiologists and their institutions to properly identify and label anesthesia-related equipment according to the conventions for each MRI scanner to ensure patient safety and prevent adverse events [18]. In the current study, although most participants reported proper labeling of anesthesia-related equipment, a significant minority reported poor labeling (Table 1). Mislabeled monitoring devices have been identified as a significant factor contributing to patient misidentification incidents, accounting for approximately 7% of such incidents across various medical practices [19]. In a previous study specifically focusing on MRI-related incidents, identification and documentation errors were found to account for 3% of the reported incidents, and they occurred at a rate of 0.011% [20]. Similarly, other studies have reported the rates of identification and labeling errors in radiology to be around 0.017% [21]. These findings highlight the importance of accurate and meticulous labeling and documentation practices in radiology and imaging departments to mitigate the risks associated with patient misidentification and ensure patient safety [17,19,20].

While general anesthesia is frequently employed, the utilization of sedation has significantly eased the assessment of pediatric patients undergoing MRI, especially for procedures that have shorter acquisition times and, thus, do not necessitate general anesthesia [18]. The current study highlights the variations in the practice of sedation administration in the MRI departments of Saudi hospitals. The present study revealed that the majority (77.4%) permitted radiologists to perform sedation. This diversity in practice is influenced by factors such as the growing demand for imaging services and the limited availability of anesthesiologists, particularly in remote areas [21]. However, it is important to recognize that the use of sedation techniques, particularly when administered by non-anesthetists, carries a potential risk of serious complications. The NCEPOD report from 2000 highlighted concerns regarding sedation-related issues in radiology within the UK [22]. The report recommended the implementation of comprehensive monitoring during vascular procedures, as well as the presence of a responsible individual other than the radiologist overseeing the process. Furthermore, the sedation protocol should undergo regular review and assessment by the anesthesia department to ensure its alignment with best practices and patient safety. This collaborative approach can help mitigate the potential risks associated with sedation administration and contribute to enhanced patient care outcomes.

This study’s findings should be interpreted within the context of certain limitations. One notable limitation is the small sample size, which may restrict the generalizability of the findings to a larger number of hospitals, including private hospitals, which were not included in the current study in Saudi Arabia. Expanding the sample size to encompass a wider range of healthcare settings would provide a more comprehensive understanding of the knowledge and attitudes about MRI safety. Moreover, it is important to acknowledge the potential for selection bias in this study. As participation in the nationwide evaluation was voluntary, hospitals with higher compliance rates may have been more inclined to participate, while those with lower compliance rates may have been underrepresented. This could introduce a bias, leading to an overestimation of overall compliance levels within the healthcare system. Additionally, the study relied on self-reporting by the participants, which could have introduced biases and inaccuracies in the data collected.

## 5. Conclusions

The current study shows a deficiency in following the guidelines and international recommendations for the safe administration of anesthesia in MRI settings in Saudi Arabia. Notably, risks associated with the absence of specialized training for MRI emergencies, communication breakdowns regarding high-risk patients, and a lack of clear policies for anesthesia in MRI units emerged as high-priority concerns. Future work is needed to identify and implement efficient strategies for enhancing anesthesia practice within Saudi Arabian MRI units. This involves embracing standardized policies, advancing training programs, and fostering improved communication among multi-disciplinary teams. These measures hold significant potential for elevating patient outcomes and elevating the overall quality of healthcare delivery.

## Figures and Tables

**Table 1 healthcare-11-02508-t001:** Anesthesia performance and safety concerns for patients in magnetic resonance imaging departments (*n* = 31).

Recommendation Guidelines	Details	*n* = 31 (%)
Anesthesia in remote sites should be provided by experienced consultants	Yes	17 (54.8%)
	No	14 (45.2%)
2. Anesthesiologists are always supported by anesthetic assistant staff while administering anesthesia in MRI units	Yes	30 (96.8%)
	No	01 (3.2%)
3. Anesthesia-related equipment is properly labeled in the MRI unit	Yes	24 (77.4%)
	No	07 (22.6%)
4. Anesthesiologists are aware of the limitations of the available monitoring equipment in the MRI unit	Yes	22 (71%)
	No	09 (29%)
5. Anesthesiologists are fully informed about how patients will be managed during MRI practice when they have a high-risk condition	Yes	23 (74.2%)
	No	08 (25.8%)
6. The MRI unit at the facility always has suction equipment available for rapid access to the patient’s airway	Yes	28 (90.3%)
	No	03 (9.7%)
7. Alternative MRI-compatible airway devices for MRI suites are always available in the facilities	Yes	27 (87.1%)
	No	04 (12.9%)

MRI, magnetic resonance imaging.

**Table 2 healthcare-11-02508-t002:** Percentage of adherence of departments of anesthesiology to safety guidelines (education and training) in MRI units (*n* = 31).

Recommendation Guidelines	Details	*n* = 31 (%)
Presence of written policies/guidelines for safety regulations to be followed while administering anesthesia in MRI units	Yes	13 (41.9%)
	No	18 (58.1%)
2. A general safety education is provided for anesthesiologists on the specific physical environment of the MRI scanner	Yes	06 (19.4%)
	No	25 (80.6%)
3. Anesthesiologists undergo specialized training to handle emergency services within MRI units, particularly when managing cardiac arrests	Yes	14 (45.2%)
	No	17 (54.8%)

MRI, magnetic resonance imaging.

**Table 3 healthcare-11-02508-t003:** Communication breakdowns between departments of anesthesiology and MRI teams when managing patients with high-risk medical conditions and implanted devices in MRI units (*n* = 31).

Recommendation Guidelines	Details	*n* = 31 (%)
MRI teams always communicate with anesthesiologists when it comes to treating patients with high-risk medical conditions and who have impaired respirators	Yes	08 (25.8%)
	No	23 (74.2%)
2. MRI teams always communicate with anesthesiologists if the patients require physiological or invasive monitors	Yes	08 (25.8%)
	No	23 (74.2%)
3. MRI teams always communicate with anesthesia units if the patients carry implanted devices	Yes	07 (22.6%)
	No	24 (77.4%)
4. MRI teams always communicate with anesthesiologists in determining whether the patients will be administered a gadolinium contrast agent	Yes	26 (83.9%)
	No	05 (16.1%)
5. MRI teams always communicate with anesthesiologists to determine whether the patients have high-risk conditions and how they will be managed	Yes	24 (77.4%)
	No	07 (22.6%)

MRI, magnetic resonance imaging.

**Table 4 healthcare-11-02508-t004:** A comparison of guidelines between university hospitals (*n* = 9) and non-university hospitals (*n* = 22).

Summary of Guidelines	University Hospitals (*n* = 9), *n* (%)	Non-University Hospitals (*n* = 22), *n* (%)	*p*-Value
General anesthesia by consultant anesthesiologists	4 (44.5%)	13 (59.1%)	0.9
2. Anesthetic assistants in MRI units	9 (100%)	21 (95.5%)	>0.9
3. MRI anesthesia equipment labeling	9 (100%)	15 (68.2%)	0.4
4. Pre-procedural notification protocols	8 (88.9%)	14 (63.6%)	0.1
5. Anesthesia unit has knowledge about monitoring equipment	8 (88.9%)	15 (68.2%)	0.9
6. Anesthesiologists informed about high-risk patient management in MRI	9 (100%)	19 (86.3%)	0.9
7. Suction equipment and MRI-compatible airway devices	5 (55.6%)	7 (31.8%)	0.8
8. Written policies/guidelines for anesthesia in MRI	6 (66.7%)	0 (0%)	0.001
9. General safety education about MRI environments	9 (100%)	5 (31.8%)	0.001
10. Specialized training for MRI emergencies, including arrests	7 (77.8%)	1 (5%)	<0.001
11. Communication about high-risk patients with respirators	3 (33.3%)	4 (22.7%)	>0.9
12. Communication about physiological or invasive monitors	5 (55.6%)	12 (54.6%)	>0.9
13. Communication about implanted devices	3 (33.3%)	8 (36.4%)	>0.9
14. Communication about gadolinium contrast agents	6 (66.7%)	16 (72.7%)	>0.9
15. Communication about managing high-risk conditions	6 (66.7%)	16 (72.7%)	>0.9

**Table 5 healthcare-11-02508-t005:** A comparison of guidelines between tertiary hospitals (*n* = 17) and non-tertiary hospitals (*n* = 14).

Summary of Guidelines	Tertiary Hospitals (*n* = 17), *n* (%)	Non-Tertiary Hospitals (*n* = 14), *n* (%)	*p*-Value
General anesthesia by consultant anesthesiologists	10 (58.8%)	7 (46.7%)	0.9
2. Anesthetic assistants in MRI units	17 (100%)	13 (92.9%)	>0.9
3. MRI anesthesia equipment labeling	14 (82.3%)	10 (71.4%)	0.9
4. Pre-procedural notification protocols	15 (88.2%)	9 (64.2%)	0.7
5. Anesthesia unit has knowledge about monitoring equipment	12 (76.5%)	11 (78.6%)	>0.9
6. Anesthesiologists informed about high-risk patient management in MRI	15 (88.2%)	9 (64.2%)	0.7
7. Suction equipment and MRI-compatible airway devices	8 (47.1%)	4 (28.6%)	0.9
8. Written policies/guidelines for anesthesia in MRI	3 (29.2%)	3 (20%)	>0.9
9. General safety education about MRI environments	9 (53%)	5 (35.7%)	0.9
10. Specialized training for MRI emergencies, including arrests	5 (18.3%)	3 (20%)	>0.9
11. Communication about high-risk patients with respiratory issues	4 (23.5%)	3 (20%)	>0.9
12. Communication about physiological or invasive monitors	8 (47.1%)	9 (64.2%)	>0.9
13. Communication about implanted devices	10 (58.8%)	7 (36.4%)	0.8
14. Communication about gadolinium contrast agents	14 (82.3%)	10 (71.4%)	0.9
15. Communication about managing high-risk conditions	15 (88.2%)	9 (64.1%)	0.7

**Table 6 healthcare-11-02508-t006:** A risk prioritization assessment for various health hazards in MRI settings based on their likelihood, impact, and risk appetite.

Malpractice from Non-Guideline Adherence	Risk Impact Category	Likelihood	Risk Appetite	PRN
General anesthesia by non-consultant anesthesiologist	4	3	2	24
2. Absence of anesthetic assistant in MRI units	2	1	3	6
3. Absence of MRI anesthesia equipment labeling	5	1	5	25
4. Lack of pre-procedural notification protocols	2	2	2	8
5. Lack of knowledge about monitoring equipment	5	2	4	40
6. Uninformed about high-risk patient management in MRI	5	1	5	25
7. Absence of suction equipment and MRI-compatible airway devices	4	1	5	20
8. No written policy	2	3	1	6
9. Absence of safety education about MRI environment	4	4	4	64
10. Absence of specialized training for MRI emergencies	5	3	5	75
11. Communication breakdown about high-risk patients	4	4	4	64
12. Communication breakdown about physiologic or invasive monitors	3	4	4	48
13. Communication breakdown about implanted devices	3	4	4	48
14. Communication breakdown about gadolinium contrast agent	4	1	4	16
15. Communication breakdown about managing high-risk conditions	4	2	4	32

Color coding of Priority Risk Number (PRN) scores (red: highest (75–125); orange: High (40–60); yellow: intermediate (24–36); light green: low (10–20); green: lowest (3–9)).

## Data Availability

The raw/processed data required to reproduce the above findings cannot be shared at this time, as the data also form part of an ongoing study.

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
