# Peer review of "Provision of Safe Anesthesia in Magnetic Resonance Environments: Degree of Compliance with International Guidelines in Saudi Arabia"

_healthcare, 2023, doi:10.3390/healthcare11182508_

Round 1

Reviewer 1 Report

Abstract: It provides a concise overview of the study's objectives, methods, and key findings. It effectively summarizes the study's main points and conclusion. However, there are a few areas where the abstract could be improved for clarity and completeness.

It mentions communication breakdown between anesthesia providers and MRI teams, particularly in patient with high-risk medical conditions and implanted devices. It would be good to provide more context on its impact. It could also benefit from a brief mention of the study limitations

Introduction: A clear and concise research objective would help readers understand the study's purpose from the outset.

Methods: Providing a brief description of the criteria selection of the facilities , including geographical distribution, hospital types or other relevant factors would give more context on the representativeness of the sample and potential biases.

Results: The section presents the findings of the study, that are completely in accord with the performed analysis.

The conclusion section highlights the importance of developing specific anesthesia guidelines, and also emphasizes the significance of effective communication and collaboration between anesthesia providers and MRI teams. It would be helpful addressing challenges that can help to anticipate and overcome potential obstacles in the improvement process.

The reading of the paper is fine, readers can easily follow the main idea. Nevertheless, the text needs a comprehensive proofreading to highlight its value. 

Author Response

Thank you for your comments and suggestions 

please check the attachment for our reply 

Reviewer 2 Report

Dear Authors.

I find the article interesting because the safety of patients must be ensured at all times and it is necessary to know the possible deficiencies existing in any place where anesthetic procedures are performed in order to be able to remedy them.

My comments by sections with respect to this manuscript are:

1.- Abstract.

Its different sections should be specified as indicated in the journal standards: Background, Methods, Results and Conclusions.

2.- Introduction.

The importance and existing problems of the research topic are adequately addressed.

Among the needs required for the anesthetic act in this context to be safe, there is an essential requirement that is not mentioned and that should be added: the need for experienced personnel (lines 74 to 77).

The paragraph detailing the objectives of the study is confusing (lines 100 to 111). It should be worded more clearly, specifying which is the main objective and which are the secondary objectives.

3.- Materials and methods.

The type of study should be specified (line 113).

The objective of this study should not be repeated because it has already been mentioned in the introduction section (lines 114 and 115).

It is indicated that the country of residence was also recorded, although no data appear in the results (lines 129 and 130).

It should be more explicit with the type of analysis to be performed, since the characteristics of the study variables are known.

The computer program used for the statistical analysis should be specified.

I do not consider that Mann-Whitney U tests are not suitable for the inferential analysis of these variables, qualitative variables, since this test is used for ordinal or numerical variables that do not follow a normal curve (lines 138 to 140).

4.- Results.

Although it goes without saying, it should be specified which of these types of hospitals are university hospitals and which are not (lines 145 to 146) since this issue is referred to later in the discussion.

It should be specified whether the hospitals that allowed radiologists to perform sedation always did it themselves or whether anesthesiologists also performed it. And if so, what is the percentage of each.

I consider that Figures 2 and 3 do not contribute anything since the data are well detailed in the text.

The paragraph "Approximately 22.6% of hospitals restricted the administration of sedation in remote locations exclusively to anaesthesiologists, while 77.4% allowed radiologists to perform sedation " is repeated (lines 167 and 168).

The paragraph "Approximately 22.6% of hospitals restricted the administration of anaesthesia services in MRI units in remote locations and to consultants, whereas 77.4% allowed anaesthesiologists of any rank to perform these services” (lines 169 to 171) is confusing. who was restricted to administer anesthesia in the 22.6% group? could only experienced anesthesiologists administer it? in the 77.4% group could radiologists and anesthesiologists with any degree of experience administer it? by what percentage?

The paragraph "and 71% agreed with the limitations of the available monitoring equipment in the MRI unit, while 29% disagreed with it " (lines 178 and 179) is unclear. Does it mean that 100% had monitoring limitations, although 71% accepted them and perform the anesthetic act?

It should be more clearly specified that it implies “full examination support in the MRI unit” for patients with implanted devices" (line 180).

I do not understand what is meant in the paragraph "Participants in this study agreed that 83.9% of MRI specialists always notified the anaesthesiologists of the need for an examination, while 16.1% disagreed " (lines 181 and 182).

Most of the sociodemographic data collected according to the methodology, such as the participant's country of residence, are not reported.

It would be interesting to break down the specific compliance with each recommendation according to the type of hospital (tertiary/non-tertiary, university/non-university) and the type of professional administering the sedation (anesthesiologist/radiologist).

There is no information on the results of inferential statistics obtained between the different variables of the study to determine whether there are statistically significant associations or differences.

5.- Discussion.

I find it adequate by providing an analysis of the results presented and establishing relationships with previous existing studies on the subject, although it could be more consistent if it took into account the results of the inferential statistics, which do not appear. Are the differences in compliance rates between tertiary and non-tertiary hospitals, or between university and non-university hospitals statistically significant, or between anesthesiologists and radiologists statistically significant?

From my point of view, it focuses too much on anesthesiologists, the professionals most prepared to administer anesthesia in any setting, and makes no reference to radiologists who also perform sedation in the hospitals in this study and who are less prepared to perform it. The performance and needs of this type of professionals in this context should also be analyzed.

6.- Conclusions.

These seem to me to be more of a continuation of the discussion, since they only indicate a series of proposals for improvement already set out previously.

The conclusions should clearly respond to the proposed objectives. For example: According to this study, there is a deficient follow-up of the guidelines and international recommendations for safe administration in MRI settings in Saudi Arabia.

7.- References.

References are appropriate to address this study topic, although they should be in accordance with journal standards.

Kind regards.

Author Response

Thank you for your comments and suggestions 

Please check the attachment for our reply 

Round 2

Reviewer 2 Report

Dear Authors.

I consider that the manuscript can be published because it has been sufficiently improved and I have no additional comments or suggestions.

Kind regards.